# Unraveling Cross-Modality Knowledge Conflicts in Large Vision-Language Models

## Abstract

Large Vision-Language Models (LVLMs) have demonstrated impressive capabilities for capturing and reasoning over multimodal inputs. However, these models are prone to parametric knowledge conflicts, which arise from inconsistencies of represented knowledge between their vision and language components. In this paper, we formally define the problem of **cross-modality parametric knowledge conflict** and present a systematic approach to detect, interpret, and mitigate them. We introduce a pipeline that identifies conflicts between visual and textual answers, showing a persistently high conflict rate across modalities in recent LVLMs regardless of the model size. We further investigate how these conflicts interfere with the inference process and propose a contrastive metric to discern the conflicting samples from the others. Building on these insights, we develop a novel dynamic contrastive decoding method that removes undesirable logits inferred from the less confident modality components based on answer confidence. For models that do not provide logits, we also introduce two prompt-based strategies to mitigate the conflicts. Our methods achieve promising improvements in accuracy on both the ViQuAE and InfoSeek datasets. Specifically, using LLaVA-34B, our proposed dynamic contrastive decoding improves an average accuracy of 2.24%.

## 1 Introduction

Large Vision-Language Models (LVLMs; OpenAI 2023; Anil et al. 2023; Liu et al. 2024) have demonstrated potent capabilities for perceiving and understanding information across different modalities. These models typically consist of a visual encoder and a large language model (LLM), aligned by a projection layer (Li et al., 2022a; Alayrac et al., 2022; Liu et al., 2024). This alignment and collaboration mechanism between language and vision components allows users to input text and images simultaneously, breeding some of the wildest applications, including retrieving information based on a combination of textual and visual queries (Karthik et al., 2023) and accomplishing complex real-world tasks with multimodal agents (Zhang & Zhang, 2023; Zheng et al., 2024).

However, the disentangled training processes and distinct learning resources leveraged by the vision and language components of an LVLM, respectively, inherently bring along inconsistencies in their learned representations, captured knowledge, as well as their influence during inference (Bartsch et al., 2023; Rabinovich et al., 2023). Given that the visual encoder and the LLM are separately trained on different datasets with distinct training objectives, their parametric knowledge across language and vision modalities is susceptible to conflicts (Wang et al., 2024), potentially leading to hallucinations (Ji et al., 2023) and inconsistencies in prediction (Chang & Bergen, 2024). As illustrated in Fig. 1, we present a conflict case from an LVLM. When asked a question about the same entity presented in two different modalities, the LVLM provides two contradictory answers. Even though the visual encoder is able to recognize the `Sydney Opera House`, the model still fails to integrate this information coherently across modalities. This phenomenon reveals a crucial challenge: the disparity between the knowledge captured by the vision and language components of LVLMs. However, there has been limited research on parametric knowledge conflicts within these models, especially concerning cross-modality conflicts. Thus, in this paper, we systematically investigate the phenomenon of **cross-modality parametric knowledge conflict** as defined in §3. We aim to address several principled research questions, as further detailed below:

*RQ1: How to detect cross-modality parametric knowledge conflicts?*

Figure 1: A conflict case of different input modalities with the same information. The conflict still happens even when the visual components recognize the Sydney Opera House.

In §4, we introduce a pipeline for detecting such conflicts using a multiple-choice question answering format focused on named entities. Specifically, we present each named entity in different modalities and pose the same question about it. The resulting answers derived from the knowledge of each modality are then compared to determine if a conflict exists. Our findings reveal a persistently high conflict rate across various model scales and architectures, indicating that increasing model size alone does not resolve these conflicts.

*RQ2: How can cross-modality parametric knowledge conflicts be interpreted, especially how they intervene the inference process?*

Given the severity of knowledge conflicts in LVLMs, this intriguing question arises. One might initially assume that such cross-modal conflicts would reduce the prediction confidence in the original answer due to conflicting parametric knowledge. However, our analyses demonstrate that confidence cannot reliably distinguish between correct and incorrect answers, necessitating a more nuanced interpretation of these conflicts. To address this issue, we propose a contrastive metric in §5 that more effectively identifies conflicting samples. This metric suggests that cross-modality knowledge conflicts actually widen the information gap embedded in the tokens.

*RQ3: What strategies can be introduced to mitigate cross-modality knowledge conflicts at inference?*

Having gained an understanding of how these conflicts affect the inference, we seek to address this question. Inspired by the strong discriminatory power of the contrastive metric, we propose a dynamic contrastive decoding method in §6. This method selectively removes undesired logits inferred from the less reliable modality based on answer confidence. Additionally, we propose two prompt-based strategies to mitigate cross-modality knowledge conflicts in cases where the model does not provide logits. Our dynamic contrastive decoding method provides more consistent improvements.

In summary, the main contributions of this paper are threefold: **1)** To the best of our knowledge, this is the first-of-its-kind work to define and study cross-modality parametric knowledge conflicts in LVLMs. **2)** We propose a practical pipeline for detecting such conflicts, along with a metric that distinguishes conflicting samples from non-conflicting ones. **3)** We introduce a dynamic contrastive decoding method to mitigate these conflicts, as well as two prompt-based strategies for resolving conflicts in closed-source models.

## 2 RELATED WORK

**Knowledge Conflict.** Knowledge conflict is a critical problem in context-specific tasks, such as machine reading comprehension (Longpre et al., 2021; Zhou et al., 2023; Wang et al., 2023a) and information extraction (Wang et al., 2022; Fang et al., 2024; Xu et al., 2022; Wang et al., 2023b;c) In the realm of LLMs, recent studies can be categorized into context-memory conflict, inter-context conflict, and intra-memory conflict (Xu et al., 2024). The context-memory conflict and the inter-context conflict are concerned mainly in the process of Retrieval Augmented Generation (RAG). They find that LLMs tend to overly rely on their own parametric memory when facing contradictory evidence (Xie et al., 2023; Wu et al., 2024). The intra-memory conflict, on the other hand, is rooted in the pre-training corpus which contains inaccurate and misleading information (Bender et al., 2021; Lin et al., 2021; Kandpal et al., 2023). The inconsistency of knowledge causes LLMs to generate outputs that are contradictory to each other when given different prompts with the same information (Elazar et al., 2022; Grosse et al., 2023), undermining their reliability. In this context, prior work has not systematically studied this problem for LVLMs, which is exactly the focus of this work.

**Robustness Issues of LVLMs.** Although LVLMs have demonstrated significant potential in understanding and reasoning over multimodal inputs, they also face several robustness challenges, including language bias (Niu et al., 2021; Zhang et al., 2024; Wang et al., 2024), hallucinations (Huang et al., 2024; Zhu et al., 2024), and the visual perception gap (Ghosh et al., 2024). Language bias refers to the tendency of LVLMs to rely on language patterns learned during LLM pretraining (Niu et al., 2021; Zhang et al., 2024; Wang et al., 2024). Hallucinations, which originate from LLMs, pertain to the discrepancies between generated contents and facts from either real-world or user inputs. (Huang et al., 2023; 2024). The visual perception gap refers to the phenomenon that the LVLMs demonstrate proficient knowledge and visual recognition abilities but fail to link their visual recognition to this knowledge (Lee et al., 2023; Ghosh et al., 2024). These issues often overlook the potential conflicts between the visual and textual components of the LVLM, which may contribute to the aforementioned challenges.

**Inference-time Intervention.** Inference-time intervention encompasses a range of techniques designed to influence the inference or generation process of LLMs (Damera Venkata & Bhattacharyya, 2022; Li et al., 2024b). These techniques either directly manipulate the logits of the generated tokens or adjust the parameters of the LLM during inference. One of the most notable strategies in this context is contrastive decoding (Li et al., 2022b; Leng et al., 2024; Zhang et al., 2024), which mitigates undesired distributions by removing them from the original distribution. Another approach involves modifying specific layers of the LLMs. For instance, ITI (Li et al., 2024b) adjusts model activation during inference by following a set of directions across several attention heads, enhancing the truthfulness of LLMs. These methods provide a means for training-free adjustments to LVLMs, significantly reducing the cost compared to readjusting model parameters.

## 3 Preliminaries

Before diving into parametric knowledge conflicts in LVLMs, we will first outline key definitions relevant to our analysis and provide an overview of the general experimental setup.

### 3.1 Definitions

To ground our analysis, we need to define 1) a typical LVLM architecture, and 2) cross-modality parametric knowledge conflicts.

**LVLM Architecture.** We focus on the general architecture that is adopted by a variety of LVLMs, including LLaVA (Liu et al., 2024), Blip (Li et al., 2023), and Qwen-VL (Bai et al., 2023). Typically, these models consist of a visual encoder $V$, a projector $F$, and a language model LM. Given a multimodal input $x_m = \{x_v, x_t\}$, where $x_v$ is the visual input and $x_t$ is the textual input, LVLM first processes $x_v$ with $V$, resulting in $p_v = V(x_v)$. Then, through the projector $F$, $p_v$ is projected into the textual embedding space: $e_v = F(p_v)$. Finally, $x_t$ is embedded into the embedding space by the embedding layer of the LM, resulting in $e_t = \text{embed}(x_t)$. The language model then generates the output by the probability $p_{\text{LM}}(y|e_v, e_t)$. So, a contemporary LVLM can be defined as $p_{\text{LM}}(y|F(V(x_v)), \text{embed}(x_t))$.

**Cross-Modality Parametric Knowledge Conflict.** Since training a large model from scratch is prohibitively costly, LVLMs typically align a vision encoder onto an existing language model. For example, LLaVA (Liu et al., 2024) aligns the pre-trained CLIP visual encoder ViT-L/14 (Radford et al., 2021) with Vicuna (Chiang et al., 2023), which have been separately trained on different data distributions, leading to potential inconsistent parametric knowledge (Grosse et al., 2023).

To elicit parametric knowledge, we propose to use answers from different modalities as the indicators of the specific parametric knowledge from each modality. Specifically, given a multimodal input $x_m = \{x_v, q\}$, where $q$ is the question regarding the entity in the image $x_v$, the output $y_m$ is generated by $p_{\text{LM}}(F(V(x_v)), \text{embed}(q_m))$, which we define as the *visual answer*. On the contrary, given a textual input $x_t = \{x_e, q\}$, where $x_e$ is the textual description of a named entity and $q$ is the question to the named entity, the output $y_t$ is generated by $p_{\text{LM}}(\text{embed}(q_t))$, which we define as the *textual answer*. If $y_m \neq y_t$, then a parametric knowledge conflict is identified.

## 3.2 EXPERIMENTAL SETUP

### 3.2.1 DATASETS CONSTRUCTION

**Original Datasets.** Following prior studies (Xie et al., 2023; Wu et al., 2024), we adopt the multiple-choice question answering (QA) as the form of evaluating cross-modality parametric knowledge conflicts. We choose two tasks of knowledge-based visual question answering about named entities:

- ViQuAE (Lerner et al., 2022) is a semi-automatically constructed dataset comprising 3.7K questions about named entities grounded in a visual context, built upon TriviaQA (Joshi et al., 2017). The named entity in the original question is replaced with an image depicting it, requiring the model to answer the question based on the visual context provided.
- InfoSeek (Chen et al., 2023) is a dataset containing 1.3M questions about over 11K visual entities, designed to evaluate the performance of LVLMs in processing visual content while acquiring relevant knowledge. The dataset is automatically constructed from templates of over 300 relations in Wikidata, ensuring a diverse set of questions.

**Multiple Choices Construction.** Given that the original datasets are free-form question answering, we synthesize distractor choices for each question. These distractor choices must be relevant to the questions to some extent but factually incorrect, to effectively evaluate the model's ability to discern the correct answers. To this end, we employ LLaMA-3-

Table 1: Statistics of the constructed multiple-choice QA dataset.

|  | ViQuAE | | InfoSeek | |
|---|---|---|---|---|
|  | Original | MCQA | Original | MCQA |
| #samples | 3,697 | 3,010 | 73,620 | 3,000 |

8B (AI@Meta, 2024) to synthesize relevant but incorrect distractor choices. The prompt used in this process is listed in Appendix Appx. §B.2. We also down-sample the InfoSeek dataset to match the sample size of the ViQuAE dataset. The statistics of the datasets are presented in Tab. 1.

### 3.2.2 EVALUATION METRICS

Since we adopt MCQA as the evaluation form, we can directly calculate the accuracy:

$$\text{Acc} = \frac{1}{N} \sum_{i=1}^{N} \mathbb{1}(y_i = \hat{y}_i), \tag{1}$$

where $N$ is the number of samples and $\hat{y}_i$ is the gold answer. Moreover, to investigate parametric knowledge conflicts, we define the inconsistency between the generated answers as **flip rate** (FR):

$$\text{FR} = \frac{1}{N} \sum_{i=1}^{N} \mathbb{1}(y_{v_i} \neq y_{t_i}), \tag{2}$$

where $y_{v_i}$ is the visual answer and $y_{t_i}$ is the textual answer. This metric indicates how many samples encounter conflicting answers between textual and visual modalities. FR only calculates cases where the textual answer contradicts with the visual answer, no matter whether the textual answer is correct or the visual answer is correct.

### 3.2.3 MODELS

Following prior works regarding LVLMs (Zhang et al., 2024; Zhu et al., 2024), we choose the LLaVA series (Li et al., 2024a) for evaluation, as they provide strong performance and a full range of model scales. Moreover, to evaluate how the architecture of LVLMs affects the phenomenon of knowledge conflicts, we adopt InstructBlip (Dai et al., 2023) and Qwen-VL (Bai et al., 2023).

## 4 DETECTING PARAMETRIC KNOWLEDGE CONFLICTS

In this section, we discuss the pipeline for detecting parametric knowledge conflicts in LVLMs and evaluate the severity of these conflicts.

### 4.1 METHOD

**Inputs.** As defined in §3.1, the visual answer is generated by asking a question about the entity presented in the image, while the textual answer is induced by replacing the image with the textual

Table 2: Results of detecting cross-modality parametric knowledge conflict. We report accuracy (Acc), recognized accuracy (R. Acc), accuracy difference ($\Delta$Acc), flip rate (FR) and the lower bound of the conflict rate (CR$_\geq$).

| Model | | ViQuAE | | | | | InfoSeek | | | | |
|---|---|---|---|---|---|---|---|---|---|---|---|
| | | Acc↑ | R. Acc↑ | $\Delta$Acc↓ | FR↓ | CR$_\geq$↓ | Acc↑ | R. Acc↑ | $\Delta$Acc↓ | FR↓ | CR$_\geq$↓ |
| LLaVA-7b | Textual | 75.65 | 78.43 | 20.32 | 41.68 | 21.36 | 52.74 | 54.55 | 27.28 | 70.13 | 42.85 |
| | Visual | 53.26 | 58.11 | | | | 22.11 | 27.27 | | | |
| LLaVA-13b | Textual | 75.65 | 69.63 | 8.37 | 36.47 | 28.10 | 56.31 | 55.41 | 19.91 | 58.44 | 38.53 |
| | Visual | 58.57 | 61.26 | | | | 31.33 | 35.50 | | | |
| LLaVA-34b | Textual | 82.46 | 82.32 | 4.37 | 24.90 | 20.53 | 66.02 | 64.07 | 15.15 | 43.72 | 28.57 |
| | Visual | 69.14 | 77.95 | | | | 44.35 | 48.92 | | | |
| InstructBlip-7b | Textual | 81.73 | 80.42 | 34.79 | 55.35 | 20.56 | 50.53 | 53.68 | 15.58 | 59.74 | 40.16 |
| | Visual | 43.09 | 45.63 | | | | 35.17 | 38.10 | | | |
| Qwen2-VL-7b | Textual | 79.30 | 78.56 | 6.19 | 28.65 | 22.46 | 63.24 | 62.77 | 2.16 | 22.51 | 20.35 |
| | Visual | 67.97 | 72.37 | | | | 61.69 | 60.61 | | | |

description of the named entity. To ensure that equal information is provided across modalities, we design distinct inputs for each, as illustrated in Fig. 1. Specifically, given a multimodal input $x_m = \{x_v, q\} \in \mathcal{D}$, where $\mathcal{D}$ is the dataset, $x_v$ is the image containing the named entity, and $q$ is the question to the named entity in $x_v$, the visual answer is generated by:

$$y_v \sim p_{\text{VLM}}(x_v, q) = p_{\text{LM}}(F(V(x_v)), \text{embed}(q)). \tag{3}$$

To generate the textual answer, we add an indicator prompt $p$ before the original question, informing the language model about the named entity in the question. $p$ is written as `This is an image of $named_entity`. Thus, the input of the textual answer becomes $x_t = p + q$. The textual answer is then generated by:

$$y_t \sim p_{\text{VLM}}(x_t) = p_{\text{LM}}(\text{embed}(x_t)). \tag{4}$$

**Irrelevant Factor Mitigation in Conflict Detection.** The results generated from the aforementioned inputs can be regarded as the elicited parametric knowledge from LVLMs. However, these answers are influenced by various other factors. For example, the visual perceiver $V$ might fail to recognize the entity in $x_v$, resulting in a random guess. These potential issues impede our ability to accurately detect cross-modality parametric knowledge conflicts. To mitigate irrelevant factors, we first instruct the LVLM to identify the entity depicted in $x_v$. If the model correctly predicts the named entity, we assume that the knowledge related to the named entity is stored in the parametric memory of $V$ and $F$, implying that any such conflict is not due to a lack of knowledge in $V$ and $F$.

### 4.2 METRIC

Despite efforts to mitigate irrelevant factors in the process of detecting cross-modality parametric knowledge conflict, certain factors remain difficult to disentangle. For instance, the visual perceiver $V$ might recognize the entity in $x_v$, but be unable to link it to the parametric knowledge within the LVLMs through the projector $F$ (Ghosh et al., 2024). Alternatively, the LVLM may be limited in its reasoning ability to relate the recognized named entity to the question. We classify these potential limitations as the *performance gap*. The performance gap leads to failures in generating the correct answer, resulting in an overall performance decline, which can be quantified by the recognized accuracy difference $\Delta$Acc = R.Acc$_{\text{textual}}$ − R.Acc$_{\text{visual}}$. Suppose that there is no conflict in the VLM, the accuracy difference between the textual and the visual answers could only be caused by this performance gap. The relationship between conflict cases and performance gap cases is illustrated in Fig. 2. Thus, we estimate the lower bound of the CR as the difference between the FR and the $\Delta$Acc. Specifically, the number of flip samples attributable to the performance gap can be calculated as $N_p = N \times \Delta$Acc, while the total number of flip samples is $N_f = N \times$ FR, where $N$ represents the total number of samples. To assess the severity of the conflicts, we calculate its lower bound as $N_{kc} \geq N_f - N_p$. Accordingly, the lower bound of the parametric knowledge conflict rate can be expressed as:

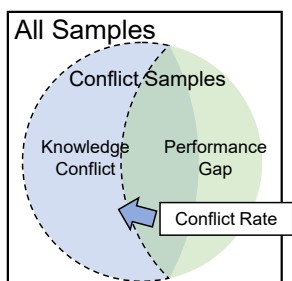

Figure 2: Relationship of conflicting samples.

$$\text{CR} = \frac{N_{kc}}{N} \geq \frac{N_f - N_p}{N} = \text{FR} - \Delta\text{Acc}. \tag{5}$$

## 4.3 ANALYSIS

We conduct experiments with LVLMs following the aforementioned procedure, and the results are presented in Tab. 2. We report the accuracy (Acc) on the complete evaluation set and the recognized accuracy (R. Acc) on the subset of the evaluation set recognized by the LVLM. Additionally, we calculate the flip rate (FR) and the conflict rate (CR) based on the recognized evaluation set.

**Performance.** For both datasets, the LLaVA-34b model demonstrates the highest accuracy for both textual and visual inputs. However, a significant performance gap exists between the textual and visual answers. The most pronounced performance gap in the LLaVA family is observed in the LLaVA-7b model, where the accuracy difference exceeds 20%. This performance gap is attributed to the cross-modality parametric knowledge conflict and the aforementioned reasons. Furthermore, there is a notable improvement in the recognized accuracy (R. Acc) across all models compared to the overall accuracy (Acc). This indicates that the models perform better on recognized entities and that the recognition process effectively mitigates potential factors influencing the final performance.

**Conflict Rate.** The flip rate (FR) decreases with increasing model size on both datasets, ranging from 55.35% to 24.90% on the ViQuAE dataset. Concurrently, the $\Delta$Acc also declines with larger model sizes, decreasing from 20.32% to 4.37% on the ViQuAE dataset. This trend likely results from the improved ability of larger models to link visual perception to parametric knowledge and their enhanced reasoning ability, rather than a reduced likelihood of parametric knowledge conflicts in larger models. When calculating the lower bound of the parametric knowledge conflict rate CR, a consistent pattern emerges across the datasets: LLaVA-7b/13b/34b exhibits values of 21.36%, 28.10%, and 20.53%, respectively. This pattern suggests that regardless of the model's scale and architecture, the likelihood of parametric knowledge conflicts remains relatively constant.

> **Key Takeaway**
>
> There is a clear trend that as the model size increases, both the FR and the $\Delta$Acc between textual and visual answers decrease. However, the lower bound of the knowledge conflict rate (CR) remains consistently high. This suggests that although scaling up models can enhance their overall performance and consistency, it does not resolve cross-modality knowledge conflicts.

## 5 INTERPRETING PARAMETRIC KNOWLEDGE CONFLICTS

The constantly large conflict rate across datasets highlights the phenomenon caused by cross-modality knowledge conflicts. In this section, we will take a closer look, through the sample-wise perspective, at how parametric knowledge in visual components, *i.e.*, the visual encoder $V$ and the projector $F$, causes cross-modality parametric knowledge conflict by intervening the inference process of the LLM. In particular, we explore how these conflicts influence answer confidence and propose a metric that can serve as an indicator of the presence of such conflicts.

### 5.1 IS PROBABILITY A RELIABLE INDICATOR OF ANSWER CORRECTNESS?

**Method.** Since the answer probability reflects the model's confidence in a given response, it is natural to consider how parametric knowledge conflicts might affect this probability. For instance, such conflicts may either reduce confidence in the original answer or introduce a more confident alternative answer. Given that embed$(x_e)$ and $F(V(x_v))$ might encapsulate different knowledge, this discrepancy can affect the probability distribution over possible answers, resulting in a shift in confidence in the final output. To investigate how cross-modality parametric knowledge conflict influences answer confidence, we design experiments to determine whether the answer confidence can serve as an indicator of conflict and whether it can suggest the correctness of the answer.

To elicit the answer probability, we calculate the textual answer probability $p_t$ and the visual answer probability $p_v$ using Eq. 3 and Eq. 4. Since we adopt MCQA as the task format, we extract the logits of the answer token, i.e. "A," "B," "C," and "D" and apply the softmax function to them. Thus, the extracted confidence can be presented as $c = \text{softmax}(\log(p[A]), \log(p[B]), \log(p[C]), \log(p[D]))$, where $p[A]$ indicates the probability of token "A," and so on. Then, we use the following strategies to understand how visual components influence the inference:

1. *Max confidence*: $\max(c_t[y_t], c_v[y_v])$, where the most confident answer is considered correct.

2. *Max confidence shift*: $\max(c_t[y_t] - c_t[y_v], c_v[y_v] - c_v[y_t])$, where $y_t$ is the textual answer and $y_v$ is the visual answer, indicating that the modality with the most significant influence on the answer is deemed the dominant modality for the question.

3. *Min variance*: $\min(\sigma(c_t[y_t]), \sigma(c_v[y_v]))$, where the answer with the least variance under disturbance is considered the final answer. We introduce disturbance through two methods: writing diverse prompts and applying the Monte Carlo dropout (Gal & Ghahramani, 2016).

**Results.** The results of three strategies are listed in Tab. 3, and the complete experimental setup is described in Appx. §B.1. From these results, it is evident that none of the strategies based on token probability reliably selects the correct answer when conflicts arise between the textual and visual answers. This suggests that: 1) *Confidence is not necessarily reduced by conflicts.* The presence of a cross-modality parametric knowledge conflict does not inherently lower the confidence level of the answer. Instead, the conflict often introduces an alternative answer

Table 3: Testing different answer correctness indicators based on answer confidence.

| Method | ViQuAE | |
| --- | --- | --- |
| | Acc | R. Acc |
| Textual Answer | 75.65 | 78.43 |
| Visual Answer | 53.26 | 58.11 |
| Max Confidence | 54.22 | 60.14 |
| Max Confidence Shift | 54.29 | 60.14 |
| Min Variance Prompt | 55.51 | 61.41 |
| Min Variance Dropout | 46.51 | 50.72 |

with higher confidence, overshadowing the original, potentially correct answer. This observation indicates that high confidence alone is not a reliable indicator of answer correctness in the presence of such conflicts. 2) *Confidence shifts are not indicative of reliability.* The results show that a greater shift in confidence between the textual and visual answers does not necessarily correlate with the reliability of the final answer. 3) *Cross-modality parametric knowledge conflict is not an uncertainty issue.* The table also reveals that methods based on variance do not contribute to the performance. Although these methods attempt to select the more stable answer by selecting the answer with minimum variance in token probability, the results show reductions in accuracy. This implies that minimizing variance does not effectively address the underlying knowledge conflicts.

## 5.2 CONTRASTIVE METRIC AS INDICATOR OF CONFLICTS

**Method.** To effectively understand how conflicting knowledge affects the inference, we utilize the concept of Contrastive Decoding (Li et al., 2022b). Its objective, which subtracts an undesired distribution from the original distribution, serves as a metric for evaluating the degree of divergence between the two distributions. Given that we are using MCQA as the task format, our focus is specifically on the distribution of the answer token, particularly the first token.

Specifically, given a multimodal input $x_m = \{x_v, q\}$, where $x_v$ is the image and $q$ is the question, and a textual input $x_t = \{x_e, q\}$, where $x_e$ is the textual description of the named entity in $x_v$, the predicted first token distribution of answers for each modality can be represented as Equations (3) and (4). The contrastive objective can then be written as:

$$\log(p_{cd}) = \log(p_v) - \log(p_t) = \log(\frac{p_{\text{VLM}}(y_v|x_v, q)}{p_{\text{VLM}}(y_t|x_e, q)}) = \log(\frac{p_{\text{LM}}(y_v|F(V(x_v)), \text{embed}(q))}{p_{\text{LM}}(y_t|\text{embed}(x_e), \text{embed}(q))}). \quad (6)$$

Ideally, if $F(V(x_v))$ and $\text{embed}(x_e)$ provide the same information for $q$, Eq. 6 should be equal to 0. However, due to the parametric knowledge conflict between the visual components and the LLM, $V(F(x_v))$ may not embed the same knowledge as $\text{embed}(x_e)$, leading to $log(p_{cd}) \not\approx 0$. Thus, $|\log(p_{cd})|$ can be interpreted as the degree of difference between $V(F(x_v))$ and $\text{embed}(x_e)$. Additionally, the contrastive decoding objective also allows us to elicit visual memories by eliminating the influence of textual knowledge. The analyses of the elicited memories are listed in Appx. §A.

**Result.** In Fig. 3, we present the distribution of the contrastive metric, specifically separating samples with consistent answers across modalities from those with conflicting answers. The figure reveals a significant disparity between the consistent and conflicting samples. Most consistent samples fall within the range of 0-0.6, while conflicting samples exhibit greater variability, with an average median of 1.46. This similar trend suggests that the extent of conflicts, as measured by the contrastive metric, is relatively consistent across different models, despite variations in model scales and architectures, implying that the cross-modality parametric knowledge conflicts are not solely dependent on the model's architecture or size but are intrinsic challenges that persist across

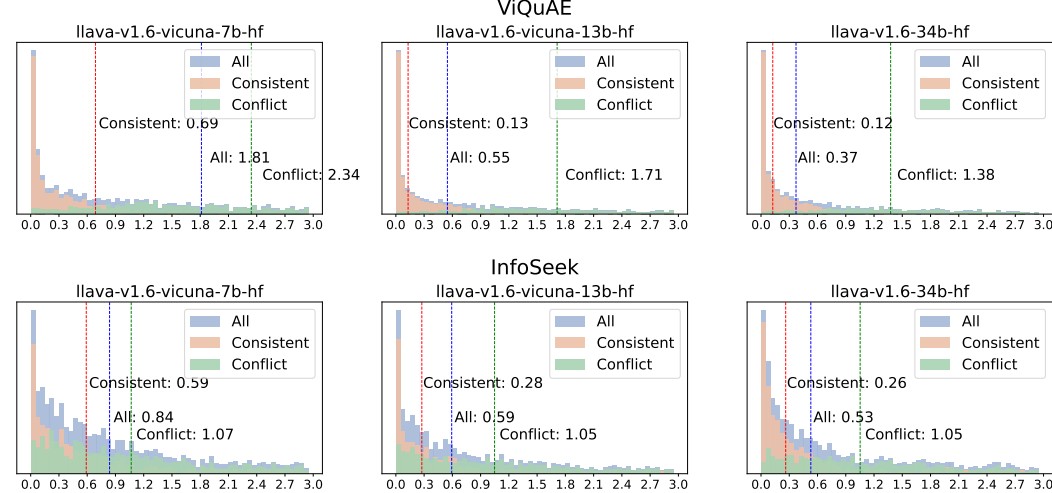

Figure 3: Distribution of the contrastive metric on all samples, samples with modality-consistent answers, and samples with modality-conflict answers. The dashed lines indicate the medians.

current training datasets. The figure also suggests that the contrastive metric is effective in distinguishing between consistent and conflicting answers. From the perspective of the contrastive metric, it quantifies the divergence between the knowledge encoded in the visual components and the LLM. Thus, the misaligned knowledge leads to the information gap embedded in the tokens of different modalities, which is ultimately presented by the conflicting answer.

> **Key Takeaway**
>
> Confidence alone is not a reliable indicator of answer correctness when confronted with conflict samples. The proposed contrastive metric effectively distinguishes conflicting samples from consistent ones, suggesting that cross-modality knowledge conflicts tend to exacerbate the information gap between tokens across different modalities, regardless of the model size.

## 6    MITIGATING PARAMETRIC KNOWLEDGE CONFLICTS AT INFERENCE TIME

Having established an understanding of cross-modality parametric knowledge conflicts, we now shift our focus to strategies for mitigating these conflicts. Since the contrastive metric has proven effective in distinguishing conflicting samples from consistent ones, we first propose a strategy that leverages the principles of contrastive decoding. Moreover, we also design an alternative approach based on prompting for models that do not provide access to logits during inference.

### 6.1    DYNAMIC CONTRASTIVE DECODING

**Method.** In an ideal application of contrastive decoding, we would have an a priori knowledge of the logits, which enables us to define the undesired logits. That is to say, to resolve cross-modality parametric knowledge conflicts, the logits from the incorrect, conflicting modality should be excluded from those of the correct modality. However, in real-world scenarios, without external validation, it is impossible to definitively determine the correctness of an answer. Therefore, we propose using the model's answer confidence as a trend for correctness, also treating it as a scaling factor for the original logits. We then apply these scaled logits to the contrastive decoding algorithm, formulating the **dynamic contrastive decoding (DCD)**. This approach adjusts the contrastive decoding objective by incorporating confidence as a dynamic factor to more accurately measure the difference in information embedded by the textual and visual components.

Specifically, given the textual answer $y_t$ with its probabilities $p_t(y_t|x_e, q)$ and the visual answer $y_v$ with its probabilities $p_v(y_v|x_v, q)$, we first calculate the confidence for each answer as follows:

$$c_t = \max(\text{softmax}(\log(p_t[\text{A}]), \log(p_t[\text{B}]), \log(p_t[\text{C}]), \log(p_t[\text{D}]))), \tag{7}$$

$$c_v = \max(\text{softmax}(\log(p_v[\text{A}]), \log(p_v[\text{B}]), \log(p_v[\text{C}]), \log(p_v[\text{D}]))), \tag{8}$$

Table 4: Results of the dynamic contrastive decoding compared to the baselines. **Bold** indicates best results and underline indicates second bests.

| Model | Method | ViQuAE | | InfoSeek | |
|---|---|---|---|---|---|
| | | Acc | R. Acc | Acc | R. Acc |
| LLaVA-7b | Textual Answer | 75.65 | 78.43 | 52.74 | 54.55 |
| | Visual Answer | 53.26 | 58.11 | 22.11 | 27.27 |
| | **DCD** | 76.49 (+0.84) | 79.51 (+1.08) | 54.90 (+2.16) | 58.87 (+4.32) |
| LLaVA-13b | Textual Answer | 75.65 | 69.63 | 56.31 | 55.41 |
| | Visual Answer | 58.57 | 61.26 | 31.33 | 35.50 |
| | **DCD** | 76.58 (+0.93) | 74.14 (+4.51) | 58.03 (+1.72) | 56.52 (+1.11) |
| LLaVA-34b | Textual Answer | 80.99 | 82.32 | 66.02 | 64.07 |
| | Visual Answer | 69.14 | 77.95 | 44.35 | 48.92 |
| | **DCD** | **83.35** (+2.36) | **85.33** (+3.01) | **68.14** (+2.12) | **67.72** (+3.65) |
| InstructBlip-7b | Textual Answer | 81.73 | 80.42 | 50.53 | 53.68 |
| | Visual Answer | 43.09 | 45.63 | 35.17 | 38.10 |
| | **DCD** | 82.47 (+0.74) | 80.59 (+0.17) | 50.53 (+0.00) | 54.38 (+0.70) |
| Qwen2-VL-7b | Textual Answer | 79.30 | 78.56 | 63.24 | 62.77 |
| | Visual Answer | 67.97 | 72.37 | 61.69 | 60.61 |
| | **DCD** | 80.76 (+1.46) | 80.59 (+2.03) | 64.30 (+1.06) | 63.34 (+0.57) |

where $p[A]$ indicates the probability for token "A," and similarly for other tokens. Next, the scaled logits are computed as $s_t = c_t \times \log(p_t)$ and $s_v = c_v \times \log(p_v)$. To assess which modality is more likely to provide the correct answer, we view the confidence as the likelihood, selecting the modality with the higher confidence. However, as discussed in §5.1, confidence alone is insufficient to determine correctness. Therefore, we subtract the scaled logits of the less confident modality from those of the more confident one. This leads to the application of contrastive decoding on the scaled logits, conditioned by the answer confidence:

$$\log(p_{cd}(y|x)) = \begin{cases} c_t \times \log(p(y_t|x_e, q)) - c_v \times \log(p(y_v|x_v, q)), & \text{if } c_t > c_v \\ c_v \times \log(p(y_v|x_v, q)) - c_t \times \log(p(y_t|x_e, q)), & \text{otherwise.} \end{cases} \quad (9)$$

**Results.** Tab. 4 presents the accuracy and the recognized accuracy for different methods across the ViQuAE and InfoSeek datasets. Across both datasets and all model sizes, DCD consistently outperforms both the textual and visual answers. For instance, in the LLaVA-7b model, DCD improves the accuracy from 75.65% to 76.49% on the ViQuAE dataset. Similarly, on the InfoSeek dataset, accuracy increases from 52.74% to 54.90%. These improvements are even more pronounced in the larger models. For example, in the LLaVA-34b model, DCD increases accuracy by 2.36% on the ViQuAE dataset and by 2.12% on InfoSeek, indicating its potential in models with larger scales.

DCD demonstrates particularly significant gains in recognized accuracy (R. Acc). For instance, on the InfoSeek dataset, the recognized accuracy for the LLaVA-34b model increases by 3.65% when using DCD compared to the textual answer. This trend is consistent across all model sizes, indicating that DCD is particularly effective in improving the performance on recognized entities. The improvement in recognized accuracy is likely due to the fact that the visual answers within the recognized set are expected to contain more relevant information than those in the unrecognized set, as the visual components have some prior knowledge of these entities. Consequently, the DCD can more effectively leverage this information to discern which option is correct.

## 6.2 PROMPTING STRATEGY

**Method.** Since not all models provide the logits of the generated contents, we propose two prompt-based improvement strategy for those models. To address cross-modality parametric knowledge conflict, we design two types of prompts and the details of these prompts are provided in Appx. §B.2.

1. *Reminder prompt.* Once a knowledge conflict is detected , the model is prompted to regenerate the answer, but this time with a reminder that highlights the presence of conflicting knowledge.

2. *Answer prompt.* Since both textual and visual answers are already generated during the detection process, this prompt asks the model to determine which one is correct.

Table 5: Results of the prompt-based strategies compared to the baselines. Since the inputs of this experiment are the same as the one of the visual answer except for the prompt, we compare them to the results of the visual answer. **Bold** indicates best results and underline indicates second bests.

| Method | ViQuAE | | InfoSeek | |
|---|---|---|---|---|
| | **Acc** | **R. Acc** | **Acc** | **R. Acc** |
| *LLaVA-7b* | | | | |
| Visual Answer | 53.26 | 58.11 | 22.11 | 27.27 |
| Reminder Prompt | 53.99 (-1.66) | 57.25 (-2.53) | 21.25(-0.86) | 27.99 (+0.72) |
| Answer Conflict Prompt | 54.58 (-1.07) | 58.51 (-1.27) | 20.23 (-1.88) | 27.39 (+0.12) |
| *LLaVA-13b* | | | | |
| Visual Answer | 58.57 | 61.26 | 31.33 | 35.50 |
| Reminder Prompt | 58.57 (+0.00) | 61.26 (+0.00) | 35.53 (+4.20) | 38.10 (+2.60) |
| Answer Conflict Prompt | 57.59 (-0.98) | 59.67 (-1.59) | 34.27 (+2.94) | 39.06 (+3.56) |
| *LLaVA-34b* | | | | |
| Visual Answer | 69.14 | 77.95 | 44.35 | 48.92 |
| Reminder Prompt | 72.99 (+3.85) | 79.28 (+1.33) | 45.15 (+0.80) | 49.62 (+0.70) |
| Answer Conflict Prompt | **73.62** (+4.48) | **79.66** (+1.71) | **52.43** (+8.08) | **53.68** (+4.76) |

**Results.** Tab. 5 presents the results of prompt-based improvements using two strategies across two datasets and different model sizes. The effectiveness of these strategies varies depending on the model size. For smaller models, both prompts negatively impact performance across both datasets, with accuracy dropping by at least 1.07% on the ViQuAE dataset and 0.86% on the InfoSeek dataset. This suggests that smaller models may struggle to handle prompts reminding them of potential knowledge conflicts, as they seem unable to discern which answer is correct. Furthermore, presenting smaller models with conflicting answers seems to introduce additional confusion, as evidenced by the more substantial accuracy declines. In contrast, larger models are more effective at processing the information provided in the prompts, demonstrating an accuracy gain of 4.48% on the ViQuAE dataset and 8.08% on the InfoSeek dataset. These results indicate that larger models are better equipped to interpret and respond to the information in the prompt, likely due to their more advanced reasoning and understanding capabilities, which enable them to determine which modality is more reliable in resolving the conflict. Overall, these findings indicate that the effectiveness of prompt-based conflict resolution strategies improves with model scale, particularly when the prompt provides the model with both conflicting answers, aiding in conflict resolution.

> **Key Takeaway**
>
> Dynamic contrastive decoding (DCD) brings universal improvements against the baselines. The performance of prompting-based strategies varies depending on the model size. Larger models are better at understanding and processing the information in the designed prompts.

## 7 CONCLUSIONS

In this paper, we introduce the concept of cross-modality parametric knowledge conflicts in LVLMs, a significant issue arising from the misalignment between visual and textual modalities. We propose a systematic approach to detect these conflicts, revealing a persistently high conflict rate across all model sizes. Our findings indicate that simply scaling up models does not resolve these conflicts, highlighting the need for targeted intervention strategies. To address these challenges, we propose the contrastive metric, which effectively identifies conflicting samples by measuring the information gap between modalities. Building on this, we introduce the dynamic contrastive decoding (DCD), which selectively removes unreliable logits to improve answer accuracy. For models without access to logits, we propose two prompt-based strategies. These approaches collectively improve model performance. On LLaVA-34B, the dynamic contrastive decoding achieves an accuracy improvement of 2.36% on the ViQuAE dataset and 2.12% on the InfoSeek dataset. Our contributions advance the understanding of cross-modality parametric knowledge conflicts in LVLMs and provide practical solutions to mitigate these conflicts, leading to more robust and accurate multimodal inference.

## ETHICS STATEMENT

Our study highlights a critical concern in recent LVLMs: the parametric memories of the vision and language components are prone to conflicts. This issue underscores the potential limitations of these models, as they may produce inconsistent or unreliable outputs if these conflicts are not properly addressed. As researchers, our goal is to mitigate these risks while maximizing the benefits.

## REPRODUCIBILITY STATEMENT

Our experiments are conducted using five open-source LVLMs to ensure reproducibility. To facilitate replication of our results, we have provided the prompts used in our experiments in Appx. §B.2. Additionally, the datasets utilized in our study are included in the supplementary materials for further reference.

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

| Question | Textual Memory | Visual Memory |
|---|---|---|
| In what city did Bruce Lee grow up? | Hong Kong. | San Francisco, California, USA. |
| George Harrison was deported from which city where the early Beatles had long term engagements, because of his youth? | George Harrison was deported from Liverpool, England because of his youth. | George Harrison of the Beatles was deported from Hamburg because he lied about his age when trying to enter Germany legally. |
| What species of fly has the Latin name calliphora vomitoria? | Calliphora vomitoria is a species of fly commonly known as the "vomiting fly" or "fruit fly." It is a small, yellowish-brown insect with a short life cycle and a strong ability to reproduce quickly. The species is widely distributed throughout the world and is often found near fruit or vegetable sources. | Calliphora vomitoria is commonly known as vomiting fly or green bottle fly. It belongs to family Calliphoridae which includes blowflies and horseflies. These insects play an essential role in decomposition processes by feeding on decaying organic matter... |
| What is the name of Bob Marley's greatest hits album? | The name of Bob Marley's greatest hits album is "Bob Marley and the Wailers: Greatest Hits." | Live Forever Volume Two by Various Artists featuring Bob Marley & The Wailers includes performances captured live during concerts throughout his career culminating with performances shortly before his passing in May of 1981. |

Table 6: Examples of elicited textual and visual memories using the contrastive decoding objective.

## A  INTERPRETING CROSS-MODALITY KNOWLEDGE CONFLICTS

The contrastive decoding objective described in §5 offers a valuable tool for examining the memory embedded within the visual components of LVLMs. Specifically, the contrastive decoding metric can be reformulated in an autoregressive form:

$$p_{cd}(y|x) = \prod_{i=1}^{n} p_{cd}(y_i|x, y_{<i}) = \prod_{i=1}^{n} \frac{p_{\text{LM}}(y_v|F(V(x_v)), \text{embed}(q), y_{<i})}{p_{\text{LM}}(y_t|\text{embed}(x_e), \text{embed}(q), y_{<i})}, \quad (10)$$

where $x$ is the inputs from both modalities and $y_{<i}$ indicates the tokens generated before step $i$. This autoregressive form of contrastive decoding metric allows us to elicit visual memory from the visual components by removing the influence of textual knowledge. We accomplish this by transforming the question into a free-form query without predefined options and then examining the elicited memory of the visual components. The examples of the elicited memories are listed in Tab. 6.

From these memories, several observations can be made:

1. **LLM is better at memorizing date and location.** This aligns intuitively with the nature of the LLM's training process, where such factual knowledge frequently appears in the text corpora. It corresponds well with the expectation that language models acquire structured knowledge from reading-based data.

2. **Visual components are better at memorizing the correlation between an entity and its names and the relationship among entities.** For example, when asked the common name for `Calliphora Vomitoria`, the LLM fails to answer correctly, while the visual answer is correct. This is likely due to the training objective of aligning visual components with the LLM, during which visual components learn entity-specific knowledge by mapping images to the language space.

Table 7: Prompt for generating false options to construct the multiple-choice question answering datasets.

Given the question and its gold answer, please generate a multiple choice version of this question. Note that the wrong choices should be relevant to the question and the gold answer should be exactly copied from what is given. You can randomly put the gold answer wherever you want. Please output as a json format: {"A": Answer A, "B": Answer B, "C": Answer C, "D": Answer D}. No further explanation or note.

Table 8: Reminder prompt to mitigate cross-modality parametric knowledge conflicts.

You are an expert at question answering. Given the question, please output the answer. No explanation and further question. Be aware that your visual memory might differ from your textual memory, causing a conflict in your knowledge.

## B EXPERIMENTAL DETAILS

### B.1 EXPERIMENTAL SETUP

**Confidence Analysis.** We will describe the experimental setup of the *Min variance* strategy in §5.1. For both settings, we sample 10 times with disturbance. For the prompt disturbance, we ask the LLaMA-3-8b (AI@Meta, 2024) to rephrase the original prompt to obtain 10 different prompts and generate the answer with each of them. For the dropout disturbance, we set the dropout rate to 0.1 and sample 10 times. Then we extract the confidence of the gold answer and calculate the variance.

### B.2 PROMPTS

The details of the prompts used in our experiments are listed here. The prompt to generate false options is in Tab. 7. The reminder prompt to mitigate knowledge conflicts is in Tab. 8. The answer conflict prompt to mitigate knowledge conflicts is in Tab. 9.

### B.3 ABLATION STUDY

We conduct experiments on the LLaVA-7b model to compare the proposed DCD and the traditional contrastive decoding method, where the latter omits the confidence scaling in Eq. 9. The results, presented in Tab. 10, indicate that the confidence scaling is effective in resolving cross-modality knowledge conflicts, which further suggests that the answer confidence encapsulates valuable information about the

Table 10: Experimental results of the overall accuracy on the ViQuAE and the InfoSeek dataset.

|      | ViQuAE | InfoSeek |
|------|--------|----------|
| CD   | 70.10  | 49.05    |
| DCD  | 76.49  | 54.90    |

relative informativeness of each modality for a given question. While confidence alone may not serve as a reliable indicator, the rich information it conveys can be leveraged to enhance overall performance.

Table 9: Answer conflict prompt to mitigate cross-modality parametric knowledge conflicts.

You are an expert at question answering. Given the question, please output the answer. No explanation and further question. Be aware that your visual memory might differ from your text memory, causing a conflict in your knowledge. Your text memory is: {textual answer} and your visual memory is: {visual answer}.

