# OpenReview forum: "Unraveling Cross-Modality Knowledge Conflicts in Large Vision-Language Models"
_ICLR.cc/2025/Conference — ICLR 2025 Conference Withdrawn Submission_

### Official Review · Reviewer_E46b · 2024-10-16

**Soundness:** 3
**Presentation:** 3
**Contribution:** 2
**Rating:** 5
**Confidence:** 4

**Summary:**

This paper addresses the issue of cross-modality parametric knowledge conflicts in Large Vision-Language Models (LVLMs). The authors identify that inconsistencies between the visual encoder and the language model components can lead to contradictory outputs when processing the same entity presented in different modalities. They propose a pipeline for detecting these conflicts, introduce a contrastive metric to interpret their impact on inference, and suggest mitigation strategies like dynamic contrastive decoding.

**Strengths:**

The paper raises and formally defines the phenomenon of cross-modality parametric knowledge conflicts in LVLMs.

The authors not only detect and quantify the conflicts but also delve into their interpretation and propose practical mitigation strategies.

The introduction of a dynamic contrastive decoding method shows promise in effectively reducing knowledge conflicts during inference without the need for additional training, which is resource-efficient.

The paper provides experimental results demonstrating the persistence of knowledge conflicts across various model scales and architectures, highlighting the significance of the problem.

**Weaknesses:**

The study evaluates the proposed methods on only two knowledge-intensive datasets. This narrow scope raises concerns about the generalizability of the findings across different domains and types of data. The paper should consider adding other evaluations like MMBench, Vibe Eval, and Mistral evals to justify that the method can generalize well.

The introduction and explanation of the contrastive metric could be more detailed. Additional clarification on how it effectively distinguishes conflicting samples would strengthen the paper.

While the paper proposes using a decoding method to solve the problem, more analysis should be given to study whether the problem comes from model training or inference. While a decoding based is lightweight, I wonder whether improving LVLM training can solve the problem too.

**Questions:**

See the weaknesses above.

(1) Can the authors provide more evaluations beyond knowledge intensive tasks?

(2) More justification about why choosing this decoding method?

(3) Is the problem solvable with improving LVLM training?

---

> ### Author Response · Authors · 2024-11-21
> **Rebuttals (1/n)**
>
> Thanks for your valuable suggestions and insightful questions!
>
> 1. W1 & Q1: The study evaluates the proposed methods on only two knowledge-intensive datasets. This narrow scope raises concerns about the generalizability of the findings across different domains and types of data. The paper should consider adding other evaluations like MMBench, Vibe Eval, and Mistral evals to justify that the method can generalize well.
>
> We appreciate the suggestions. Our study focuses on the cross-modality knowledge conflicts in VLMs.
> We have looked into your suggested datasets. Please understand that these general benchmarks do not provide data to evaluate cross-modality knowledge conflict. Therefore, we have to use two knowledge-intensive dataset, i.e., the ViQuAE dataset and the InfoSeek dataset, to facilitate the study of this problem.
>
> ---
>
> 2. W2: The introduction and explanation of the contrastive metric could be more detailed. Additional clarification on how it effectively distinguishes conflicting samples would strengthen the paper.
>
> The contrastive metric quantifies the differences between the log probabilities of predicted tokens generated by the visual and textual modalities.
> This difference serves as a metric for the extent of conflicting knowledge between the two modalities.
> We included an overall figure across datasets and model scales to prove its effectiveness.
> From the figure, we can observe a large difference of the contrastive metric between the consistent samples and the conflicting samples.
>
> ---
>
> 3. Q2: More justification about why choosing this decoding method?
>
> We have the following three reasons to do so.
>
> First, the contrastive decoding method can effectively eliminate undesired logits from the original logits, thereby reducing the interference caused by conflicting parametric knowledge.
>
> Second, since the detection process generates logits as its output, it is a natural step to work directly with these logits to refine the decoding process.
>
> Finally, DCD dynamically adjusts the decoding process, ensuring that the final output is primarily influenced by the modality with higher confidence.
> This is to utilize the nature of the confidence that higher confidence from one modality may indicate that the modality could contain more relevant information than the other.
> At the same time, it leaves room for the other modality to contribute more significantly when its influence is necessary.
>
> ---
>
> 4. Q3: While the paper proposes using a decoding method to solve the problem, more analysis should be given to study whether the problem comes from model training or inference. While a decoding based is lightweight, I wonder whether improving LVLM training can solve the problem too.
>
> Thank you for your valuable suggestion. We agree that cross-modality knowledge conflicts may originate during the training phase. These conflicts often result from the separate training of visual encoders and language models on distinct datasets with differing objectives, leading to misaligned representations between the modalities. And our study focuses on detecting and interpreting these cross-modality knowledge conflicts during inference time. Thus, we did not focus on the training-time solutions. Discussing further, improving the training process could potentially address these conflicts. For instance, joint training of visual and language components on unified datasets with shared objectives might foster better alignment between modalities.

---

> > ### Author Response · Authors · 2024-11-26
> >
> > Hi Reviewer E46b,
> >
> > Sincerely thanks again for your time and efforts in reviewing our submission.
> >
> > As the discussion period is ending soon, we wonder if there's any possibility that you can consider having a look at our reply. We would like to hear your feedback. If you have any other questions, please feel free to follow up anytime. We are committed to making any necessary revisions to further improve our work.

---

> > > ### Comment · Reviewer_E46b · 2024-11-26
> > > **Thanks for the rebuttal**
> > >
> > > Thanks for your detailed responses. However, I think it is worth justifying that solving this knowledge conflict issue could improve VLM in general since some general benchmarks also contain subsets to evaluate this. Some justifications from more general datasets are useful too.

---

> > > > ### Author Response · Authors · 2024-11-27
> > > >
> > > > Thanks for your response.
> > > >
> > > > We have carefully considered your recommended datasets. However, these datasets primarily consider the visual understanding ability of the VLMs, which is not the scope of our work. And the subsets that are related to the knowledge ability of the VLMs are subsets of our evaluated datasets. For example, in MMBench, the related subsets are “Social Relation”, “Physical Relation”, “Physical Property”, and “Natural Relation”, **which are already included** in these two Wiki-based evaluation datasets. Moreover, we are trying to detect and solve the conflicts **during inference time**, but these general datasets are intended to evaluate the overall ability of a trained VLM, which is beyond the scope of our work. Thus, we kindly suggest that the focus on the knowledge-intensive datasets is necessary and sufficient to justify the integrity of both the analysis and the proposed method.
> > > >
> > > > We would like to clarify that our scope is in the cross-modality parametric knowledge conflicts in VLMs during inference time and we detect, interpret, and mitigate knowledge conflicts within this scope. Following prior works of this scope in LLMs [1,2], they also focus on analyzing and evaluating on the knowledge-intensive datasets.
> > > >
> > > >
> > > > [1] Jian Xie,et al. Adaptive chameleon or stubborn sloth: Revealing the behavior of large language models in knowledge conflicts. ICLR 2024 spotlight
> > > >
> > > > [2] Nikhil Kandpal, et al. Large language models struggle to learn long-tail knowledge. International Conference on Machine Learning. ICML 2023.

---

### Official Review · Reviewer_pyDW · 2024-11-01

**Soundness:** 3
**Presentation:** 3
**Contribution:** 3
**Rating:** 6
**Confidence:** 4

**Summary:**

This paper addresses the issue of parametric knowledge conflicts within LVLMs. It defines this cross-modality conflict problem and develop a systematic pipeline to detect, interpret, and mitigate such conflicts, revealing a consistently high conflict rate across modelse. They introduce a contrastive metric to differentiate conflicting samples and propose a dynamic contrastive decoding method that selectively removes low-confidence logits, which improves model accuracy.

**Strengths:**

- The paper introduces and formally defines a critical, underexplored issue in LVLMs, cross-modality parametric knowledge conflict, and its implications for model inference.
- A well-structured pipeline for conflict detection, interpretation, and mitigation provides a comprehensive framework for handling multimodal inconsistencies.

**Weaknesses:**

This work is compelling, as multimodal knowledge plays a vital role in advancing LVLMs. However, a fundamental challenge remains: for LVLMs, the core focus is still the generation task. The approach in this article appears limited to classification and token generation, focusing mainly on calibrating output probability. Addressing this challenge more directly is crucial for the current stage of LVLM development.

In particular, you may need to consider Image Caption or CoT Reasoning tasks, like Coco, ScienceQA, M3CoT.

**Questions:**

- Adding discussion of knowledge conflicts on how to multimodally generate tasks will be much helphul for the community.

---

> ### Author Response · Authors · 2024-11-21
> **Rebuttals (1/n)**
>
> Thanks for your valuable suggestions and insightful questions!
>
> 1. W1: In particular, you may need to consider Image Caption or CoT Reasoning tasks, like Coco, ScienceQA, M3CoT.
>
> We appreciate the suggestions. We have looked into your suggested dataset. Please understand that these datasets do not fit the scope of our study. We are studying the cross-modality knowledge conflicts between the language and vision components. Thus, we want the task to require less on other abilities except for knowledge. These datasets often require the model to focus on some details of the given image, or require other visual abilities, which further widen the performance gap defined in our paper.
>
> ---
>
> 2. Q1: Add discussion of knowledge conflicts on how to multimodally generate tasks.
>
> For the generation task, we have already included some cases of the generated answers in the appendix.
> In Appendix A, we present several examples of conflicting generated contents given the questions.
> These cases suggest that conflict still happens when given inputs from different modalities.
> We also sample 200 cases and manually calculate the flip rate of the generated contents.
> The results show that 40 cases (20%) exist of knowledge conflicts in the generation form.

---

> > ### Comment · Reviewer_pyDW · 2024-11-22
> >
> > Thanks for your reply. For W1, I think Limitation or related discussions should be added to future versions to make the paper more complete.
> >
> > Other than that, I have no other questions.

---

> > > ### Author Response · Authors · 2024-11-22
> > >
> > > Thanks for your feedback!
> > >
> > > We are happy to address your concerns.
> > > In the next version, we will include some discussions on the scope of our work.

---

### Official Review · Reviewer_G8Cf · 2024-11-07

**Soundness:** 1
**Presentation:** 3
**Contribution:** 1
**Rating:** 3
**Confidence:** 3

**Summary:**

This paper studies the phenomenon of cross-modality parametric knowledge conflict in vision language models. Specifically, when a VLM provides different answers with and and without the vision modality (with effective textual substitute for visual input), those instances are considered as cross-modality knowledge conflicts. To study this scenario, the authors construct a multiple-choice question answering dataset from two datasets: ViQUAE and InfoSeek where the questions are based on named entities presented in the accompanying image. They first extract answers based on both image and text input (visual accuracy), and then extract answers from the VLM with textual input only (textual accuracy). Their exploration shows that there can be large gaps between the textual accuracy and visual accuracy of contemporary VLMs. The accuracy on the subset of data where the VLM succeeds at recognizing the named entity is used for further evaluations. The QA accuracy gap on this subset can be attributed to knowledge conflicts and performance gaps, according to the authors. Performance gaps refer to the inability of the model to connect the visual encodings to the underlying parametric knowledge in LLMs in order to answer the question correctly. The conflict rate is calculated as the difference between flip rate (cross-modality inconsistency) and performance gap (calculated as diff. Of textual and visual acc. On the subset). Further exploration shows that confidence measured using logits is not a good indicator of instances with parametric knowledge conflict. However, a contrastive interpretation of the logits serves as a good indicator of instances where the modalities lead to knowledge conflict. Further, this paper proposes a dynamic contrastive decoding method and prompting method to mitigate cross-modality knowledge conflicts and shows significant improvements in QA accuracy for VLMs.

**Strengths:**

**Well-written paper**: This paper is well-written with a clear motivation for exploration on this topic, well-structured sections and flow of contents. Experimental results are conveyed clearly and hypothesis are discussed extensively in the main text.


**Important Topic**: Topics related to parametric knowledge are always an important research topics, with those pertaining to VLMs becoming even more important as VLMs become mainstream in day-to-day use.

**Weaknesses:**

**Soundness of the cross-modality parametric knowledge conflict explanation**:

The authors outline an explanation of the cross-modality knowledge conflict in Sec. 4.2. Following the provided explanations, these are the four categories of samples present in the evaluation set of recognized accuracy (also represented in Fig. 2):

1. Correct Text, Correct Visual
2. Correct Text, Incorrect Visual
3. Incorrect Text, Correct Visual
4. Incorrect Text, Incorrect Visual

According to the provided definitions in 4.1,

Flip Rate = size(2) + size(3)

$\Delta$Acc. = size(1) + size(2) - size(1) - size(3) = size(2) - size(3)

So, conflict rate = Flip Rate - $\Delta$Acc. = size(3) * 2

Please correct me if my understanding seems wrong. This seems like an inadequate measure for the cross-modality conflict since it only captures the size of category 3. Specifically, I think that the samples that are being attributed to performance gap (primarily category 2) should also be counted towards conflict since at least a part of this category implies that the visual-language knowledge is at odds with the language-only knowledge.

**Dynamic Decoding Method**: While this method does achieve significant improvements, this method lacks practical use because it requires that the user knows the named entity being shown via image to the VLM. If the user already has this information, then it would make more sense to simply pose this question to a capable LLM.

**Limited Generalizability of Results**: The results presented in this paper are specific to MCQ questions only. The MCQ format does make it easy to study this phenomena, however, it may not generalize to open-domain natural language answers which are the dominant format of interaction with current LLMs and VLMs.

**Questions:**

- The main motivation of this paper is to study conflicts between the LLM knowledge and VLM knowledge about the same topic. However, I am not completely convinced by the term 'conflict' for this scenario because the visual modality merely adds new knowledge to the LLM. Any conflict that arises from this addition can be solely because the training process could not bridge the gap between visual modality and language modality correctly, which has been termed as performance gap in this paper. What other reasons could there possibly be for this conflict, which is being captured in the Conflict rate? I would appreciate more discussion on this point in the rebuttal. Apologies if I have missed the point of this paper.

- How exactly is $\Delta$ accuracy calculate? Is it calculated based on the accuracies on the complete evaluation set? If yes, it might change my interpretation of the Conflict Rate outlined under Weaknesses.

- Suggestion to also report $\Delta$ accuracy in the table for easier interpretation of the results.

---

> ### Author Response · Authors · 2024-11-21
> **Rebuttals (1/n)**
>
> Thanks for your valuable suggestions and insightful questions!
>
> 1. W1&Q2: Soundness of the cross-modality parametric knowledge conflict explanation and how is $\Delta$Acc calculated?
>
> Thank you for your detailed analysis and thoughtful question.
>
> First, to answer your question about how $\Delta$Acc is calculated, the $\Delta$Acc is calculated on the recognized evaluation set.
> However, there is a misunderstanding in your calculation.
> As defined in Section 3, the Flip Rate (FR) does not consider the correctness of the answers.
> This means that when both the textual and visual answers are incorrect but inconsistent with each other (e.g., the gold answer is A, the textual answer is B, and the visual answer is C), they still contribute to the FR. Consequently, the FR includes some cases classified as Type 4 in your analysis. We have clarified this point in the revised version of our paper.
>
> Second, regarding the intuition behind calculating the lower bound of the Conflict Rate (CR), Section 4 aims to detect cross-modality knowledge conflicts by examining the consistency of answers across different modalities.
> Due to the complexity and entanglement of various factors in the generation process, it is challenging to precisely determine which inconsistencies are caused specifically by cross-modality knowledge conflicts.
> Hence, in this section, we first filter out those samples where VLMs cannot recognize it (lack of knowledge) and then eliminate the influence of other factors by subtracting the $\Delta$Acc from the FR, which estimates the lower bound of the conflict rate because some of the $\Delta$Acc may be caused by the cross-modality knowledge conflicts, as suggested by our Figure 2.
> These two steps approximately estimate the lower bound of the conflict rate by disentangling two critical factors (lack of knowledge and performance gap) from the generation process.
> It ensures that our estimation of the conflict rate is less affected by unrelated factors.
>
> Thus, in this context, it’s important to clarify that the classification of the four types of cases does not fully align with the objective of this section.
> Specifically, this classification only counts whether the answers are correct given textual and visual inputs, not considering why the answer is flipped.
> Some are caused by cross-modality knowledge conflicts, while others may attribute to the model’s poor reasoning ability.
> Therefore, calculating the lower bound of CR is to estimate how many conflicting cases are likely to be caused by knowledge conflicts.
> For example, in a scenario where no cross-modality knowledge conflict exists, after confirming that the VLM can recognize the entity in the input image, the accuracy difference between the textual and the visual answer could only be caused by the performance gap defined in our paper. Thus, subtracting the accuracy difference from the flip rate ensures that these factors are eliminated.
>
> Finally, to go beyond estimating the lower bound of the conflict rate, we introduced a contrastive metric in Section 5. This metric provides a more precise quantification of the extent of cross-modality knowledge conflicts, complementing the lower-bound analysis in Section 4.
>
>
> Thanks again for your insightful analysis. We have clarified that we are reporting the lower bound of the conflict rate in the table and added the detailed explanation of the metrics in our revised version.

---

> > ### Author Response · Authors · 2024-11-21
> > **Rebuttals (2/n)**
> >
> > 2. Q1: The main motivation of this paper is to study conflicts between the LLM knowledge and VLM knowledge about the same topic. However, I am not completely convinced by the term 'conflict' for this scenario because the visual modality merely adds new knowledge to the LLM. Any conflict that arises from this addition can be solely because the training process could not bridge the gap between visual modality and language modality correctly, which has been termed as performance gap in this paper. What other reasons could there possibly be for this conflict, which is being captured in the Conflict rate? I would appreciate more discussion on this point in the rebuttal. Apologies if I have missed the point of this paper.
> >
> > Thank you for your insightful question. This is indeed an important topic, and we are glad to have the opportunity to address it. We would like to clarify the motivation behind our study and address your concerns.
> >
> > First, we are studying the conflicts between the language model and the visual components in a single VLM.
> > These conflicts arise because the language and vision components are typically trained separately, often on different datasets with distinct training objectives.
> > This distinction makes them inherently prone to inconsistencies in their internal knowledge.
> >
> > We believe the studied problem is important and timely since current VLM training heavily relies on aligning one modality to the LLM, without considering the alignment of their respective internal knowledge.
> > In this context, we definitely cannot jump to the conclusion that ``the visual modality merely adds new knowledge to the LLM’’.
> > Otherwise, there would be no instances of type (2) Correct Text, Incorrect Visual.
> > Our experiments reveal that such cases do occur, indicating that conflicts exist even when the visual modality successfully recognizes the entity in the presented image.
> > These conflicts suggest a misalignment of parametric knowledge between the modalities, not simply the addition of new ones.
> >
> > Second, it would be overly simplistic to conclude that any conflict that arises from this addition can be solely because the training process could not bridge the gap between visual modality and language modality correctly.
> > There are lots of intriguing misalignment phenomena in VLMs, including hallucinations in the visual modality (hallucinate about objects not existing in the visual input) [1,2] and the inability to extract spatial information [3,4].
> > The former one may stem from the repeatedly-appearing object pairs in the training set, while the latter suggests that current VLM training should take spatial information into consideration.
> > These examples demonstrate that the issue extends beyond a lack of proper alignment between the two modalities; rather, a comprehensive and in-depth analysis is required to pinpoint the root causes.
> >
> > Our study highlights the need for comprehensive alignment of the parametric knowledge between visual and language modalities.
> > Current training paradigms often focus on aligning the modalities at the output level without considering their internal knowledge alignment.
> > The conflict cases, conflict rate, and the contrastive metric presented in our work comprehensively assess this misalignment, emphasizing that resolving these conflicts requires a deeper understanding of their underlying causes.
> >
> > We hope this explanation clarifies our motivation and our focus.
> > If you have further questions or suggestions, please let us know and we are happy to discuss more about it.
> >
> > [1] Zhou, Yiyang, et al. Analyzing and mitigating object hallucination in large vision-language models. ICLR 2024.
> >
> > [2] Li, Yifan, et al. Evaluating object hallucination in large vision-language models. EMNLP 2023.
> >
> > [3] Kamath, Amita, Jack Hessel, and Kai-Wei Chang. What's" up" with vision-language models? Investigating their struggle with spatial reasoning. EMNLP 2023.
> >
> > [4] Wang, Jiayu, et al. Is a picture worth a thousand words? delving into spatial reasoning for vision language models. Neurips 2024.

---

> > > ### Author Response · Authors · 2024-11-21
> > > **Rebuttals (3/n)**
> > >
> > > 3. W2: Dynamic Decoding Method: this method lacks practical use because it requires that the user knows the named entity being shown via image to the VLM. If the user already has this information, then it would make more sense to simply pose this question to a capable LLM.
> > >
> > > DCD is proposed to resolve knowledge conflicts, which is also detected by two distinct processes.
> > > These processes involve different formats of input information from two modalities.
> > > In scenarios where the user does not initially know the named entity, the VLM can first recognize the entity from the image.
> > > Once the entity is recognized, the user can pose the question again to the VLM, incorporating the identified named entity, as outlined in the pipeline proposed in Section 4.
> > > This workflow demonstrates how DCD can be applied when starting with incomplete information.
> > >
> > > Moreover, although it is reasonable to consult a more capable LLM if the named entity is already known to users, DCD still outperforms the performance of the textual answers, under the context of the same VLM.
> > > This suggests that given the same model, DCD outperforms the pipeline of recognizing named entities then generating textual answers.
> > >
> > > In general, DCD is not intended to replace standalone LLMs but rather to maximize the utility of VLMs by addressing cross-modality conflicts.
> > >
> > > ---
> > >
> > > 4. W3: Limited Generalizability of Results: only evaluating on MCQ.
> > >
> > > The reason for choosing the MCQ form is for the sake of feasibility to evaluate, and further making the analytical results in this study more verifiable. On the contrary, evaluating generative QA still faces an unresolved challenge for automated evaluation and may often still require the help of human effort.
> > > Human evaluation requires large amounts of human labors, while the automated evaluation introduces bias and uncertainty, potentially compromising the integrity of the evaluation [5].
> > > Thus, to ensure the integrity of the evaluation, we adopt MCQ form.
> > >
> > > As for your concern of the performance of the open-domain question, we actually adopted the detection pipeline on the generation process for open-domain questions.
> > > In Appendix A, we present some cases of the generated contents.
> > > We include examples of generated content, demonstrating that cross-modality conflicts persist in open-domain answers.
> > > Moreover, we conduct a manual evaluation of 200 sampled cases of LLaVA-7b on the ViQuAE dataset, calculating the flip rate for the generated answers.
> > > This analysis reveals that 40 cases (20%) exhibited knowledge conflicts, confirming the presence of conflicts even in open-domain formats.
> > >
> > >
> > >
> > > [5] Stureborg, Rickard, Dimitris Alikaniotis, and Yoshi Suhara. Large language models are inconsistent and biased evaluators. ACL 2024.
> > >
> > >
> > >
> > > ---
> > >
> > >
> > > 5. Q3: Suggestion to also report  $\Delta$ accuracy in the table for easier interpretation of the results.
> > >
> > > This is a good suggestion. We have added the $\Delta$Acc in the table of our revised version.

---

> > > > ### Author Response · Authors · 2024-11-26
> > > >
> > > > Hi Reviewer G8Cf,
> > > >
> > > > Sincerely thanks again for your time and efforts in reviewing our submission.
> > > >
> > > > As the discussion period is ending soon, we wonder if there's any possibility that you can consider having a look at our reply. We would like to hear your feedback. If you have any other questions, please feel free to follow up anytime. We are committed to making any necessary revisions to further improve our work.

---

### Official Review · Reviewer_TP54 · 2024-11-10

**Soundness:** 2
**Presentation:** 3
**Contribution:** 2
**Rating:** 5
**Confidence:** 4

**Summary:**

This work introduces the concept of knowledge conflict in vision-language models (VLMs) and designs a pipeline to detect such conflicts, tested across various VLMs. Additionally, it proposes a decoding method to mitigate the impact of these conflicts through denoising during the decoding process, thereby enhancing the performance of VLMs on visual question-answering (VQA) tasks.

**Strengths:**

1. The study addresses a generally prevalent and practically valuable problem and proposes a new perspective on the causes of hallucinations in vision-language models.

2. The presentation is clear and comprehensible, with intuitive and illustrative figures effectively conveying the proposed ideas.

3. The experiments are thorough and comprehensive, covering a variety of model sizes and architectures.

4. The proposed method demonstrates strong performance, resulting in significant improvements over baseline models.

**Weaknesses:**

1. The definition of the problem is somewhat unclear, with certain edge cases omitted, which could pose potential issues (see Question 3).

2. Some result analyses are inconsistent with the quantitative data presented in the tables (see Questions 4 and 5).

3. The motivation behind the proposed method is not sufficiently explained, particularly regarding the rationale for its design (see Questions 6 and 7).

**Questions:**

1. Why was an LLM (e.g., LLaMA) used instead of a VLM to generate incorrect distractor choices during dataset construction? The prompts in the appendix indicate that the LLM generates incorrect yet relevant choices. However, as LLaMA is a single-modality model without image input capabilities, it implies the model does not have access to the complete original question when synthesizing relevant choices.

2. What is the rationale behind down-sampling the InfoSeek dataset to match the size of the ViQuAE dataset? Comparisons should focus on models, not datasets.

3. Does the flip rate account for cases where both the visual and text answers are incorrect (but not equal)? For subsets entirely consisting of such cases, N_p=N×0=0 and N_f=N×1=N resulting in CR=1. This scenario is more likely attributable to a lack of knowledge or weak reasoning abilities in the LLM rather than knowledge conflicts.

4. In Table 2, why is the R.Acc lower than the Acc for the Qwen2-VL-7b model on the InfoSeek dataset? Could the authors provide a possible explanation for this observation?

5. In Section 4.3, the authors claim that the knowledge conflict rate (CR) is consistent across datasets, regardless of model scale. However, Table 2 shows that for the InfoSeek dataset, as the LLaVA model size increases, the CR decreases, with the reduction becoming more pronounced with larger models. This observation appears to contradict the authors' claim.

6. What is the meaning of the proposed contrastive metric? Section 5.2 states that the contrastive metric is designed to differentiate between consistent and conflicting answers. However, Equation 6, which calculates the metric, uses the logits of both the text and visual answers. If the logits for both approaches are already available, why not directly decode the final answers and determine conflicts by checking their equivalence?

7. In Section 5.1, the authors mention that confidence is unreliable for indicating the more reliable modality. Yet, in Section 6.1, confidence is used as a parameter to scale the logits, followed by a subtraction operation (Equation 7). Could the authors elaborate on why confidence is assumed to be reliable in this context and the significance of this subtraction?

8. The explanation of why larger models benefit more from the prompting strategy is unconvincing. The assumption that incorrect predictions stem from cross-modality knowledge conflicts contrasts with the demonstration that such conflicts exist across various model scales. If these conflicts consistently occur across modalities, what enables larger models to implicitly discern which modality's knowledge is more reliable?

---

> ### Author Response · Authors · 2024-11-21
> **Rebuttals (1/n)**
>
> Thanks for your valuable suggestions and insightful questions!
>
> 1. Q1: Why was an LLM used instead of a VLM to generate incorrect distractor choices during dataset construction?
>
> We appreciate your question regarding whether the model has access to the complete original question when synthesizing relevant choices. To clarify, the LLM is provided with the named entity from the question during the distractor generation process. This ensures the model to have the necessary context to create plausible options.
> Choosing LLMs is due to the fact that they are trained on extensive and diverse textual data, equipping them to generate contextually relevant and semantically appropriate distractors. Their proficiency in language processing ensures that the distractors are not only linguistically coherent but also semantically related to the correct answer. As a result, the generated multiple-choice questions maintain high quality and effectively serve their intended purpose.
>
> ---
>
> 2. Q2: What is the rationale behind down-sampling the InfoSeek dataset to match the size of the ViQuAE dataset? Comparisons should focus on models, not datasets.
>
>
> The original InfoSeek is over 70k, which will take more than 20 hours for us to evaluate one 7b model, let alone a 13b or 34b model.
> Thus, for the sake of efficiency and budget consideration, we down-sampled the InfoSeek dataset, also to match the size of the ViQuAE dataset.
>
> ---
>
> 3. Q3: Does the flip rate account for cases where both the visual and text answers are incorrect (but not equal)? For subsets entirely consisting of such cases, N_p=N×0=0 and N_f=N×1=N resulting in CR=1. This scenario is more likely attributable to a lack of knowledge or weak reasoning abilities in the LLM rather than knowledge conflicts.
>
>
> Yes, the flip rate does account for cases where both the visual and text answers are incorrect but not equal.
>
> Exactly as you mentioned, in order to amply assess the performance degradation attributed to cross-modality knowledge conflict instead of the VLM’s lacking of knowledge or other factors, we first filter out those samples that VLM cannot recognize the named entity in it (a lack of visual knowledge) and then minus the delta accuracy from the flip rate (performance gap, as we suggested, also includes weak reasoning abilities). Through these steps, the computed conflict rate should reasonably reflect knowledge conflicts rather than other factors like knowledge deficits or reasoning weaknesses.
> Additionally, to quantitatively interpret knowledge conflicts, we propose the contrastive metric in Section 5, which offers a nuanced interpretation of these conflicts.
>
> ---
>
> 4. Q4. In Table 2, why is the R.Acc lower than the Acc for the Qwen2-VL-7b model on the InfoSeek dataset? Could the authors provide a possible explanation for this observation?
>
> Thank you for pointing out this intriguing observation.
>
> Intuitively, one would expect the R.Acc to be higher than Acc, as recognizing the entity typically indicates that the VLM possesses some knowledge about it. However, if the visual components contain incorrect knowledge, it will also cause performance drop.
> In the case where Qwen2-VL-7b is evaluated on the InfoSeek dataset, we believe that the visual components may contain knowledge that is inconsistent with the language model's parametric knowledge. This misalignment between the modalities could result in a lower R.Acc, as the recognized entity might lead to incorrect answers rather than improving performance.
> In general, issues like this highlight the necessity of checking the consistency of the language and vision components of VLMs.
>
> ---
>
> 5. Q5. In Section 4.3, the authors claim that the knowledge conflict rate (CR) is consistent across datasets, regardless of model scale. However, Table 2 shows that for the InfoSeek dataset, as the LLaVA model size increases, the CR decreases, with the reduction becoming more pronounced with larger models. This observation appears to contradict the authors' claim.
>
> While it is true that CR decreases as the LLaVA model size increases, we note that the lower bound of the CR remains consistently high across datasets.
> This suggests that scaling up models enhances their cross-modality competency and overall performance, but does not entirely resolve cross-modality knowledge conflicts.
> Therefore, despite a reduction in CR with larger models, a significant level of conflicts persists, indicating that model scaling alone is insufficient to eliminate these conflicts.
>
> Moreover, this conclusion is supported by the contrastive metric analysis presented in Figure 3.
> The consistently significant gap in the contrastive metric between consistent and conflicting samples demonstrates that cross-modality knowledge conflicts are a persistent issue across datasets, irrespective of the model scale.
> This further supports our conclusion that model scaling alone is insufficient to address these conflicts comprehensively.

---

> > ### Author Response · Authors · 2024-11-21
> > **Rebuttals (2/n)**
> >
> > 6. Q6. What is the meaning of the proposed contrastive metric? Section 5.2 states that the contrastive metric is designed to differentiate between consistent and conflicting answers. However, Equation 6, which calculates the metric, uses the logits of both the text and visual answers. If the logits for both approaches are already available, why not directly decode the final answers and determine conflicts by checking their equivalence?
> >
> >
> > The proposed contrastive metric is designed to provide a quantitative measure of cross-modality knowledge conflicts at the sample level, going beyond merely classifying specific instances as causing conflict or not.
> >
> > In Section 4, we approximate the lower bound of the CR by examining the overall performance on the entire dataset, and, in Section 5, we aim to quantify it through a sample-wise way.
> > The proposed contrastive metric serves as 1) a metric to check the extent of differences of the textual and visual knowledge for a given topic, and 2) a metric to quantify the overall differences of the textual and visual knowledge across the entire dataset. While directly decoding answers and checking for equivalence can classify individual cases as consistent or conflicting, this binary approach does not capture the extent of the conflict.
> >
> > We hope this explanation clarifies the purpose of the contrastive metric.
> >
> > ---
> >
> > 7. Q7. In Section 5.1, the authors mention that confidence is unreliable for indicating the more reliable modality. Yet, in Section 6.1, confidence is used as a parameter to scale the logits, followed by a subtraction operation (Equation 7). Could the authors elaborate on why confidence is assumed to be reliable in this context and the significance of this subtraction?
> >
> >
> > For the **usage of confidence**, the key distinction lies in the application:
> > - Section 5.1: Confidence is assessed in isolation, where confidence is the only standard to determine which modality is more reliable. So, in this context, confidence alone is not a good enough indicator.
> > - Section 6.1: DCD employs a contrastive approach. Confidence is used as a scaling factor for logits instead of the sole determinant of modality reliability. Then the contrastive part eliminates the undesired logits from the scaled logits. Hence, in this context, confidence serves as a reference metric, complementing other mechanisms in determining modality reliability.
> >
> > Therefore, the intuition here is that while confidence alone cannot resolve knowledge conflicts, it provides valuable information about the extent to which the model possesses relevant knowledge and how much that knowledge contributes to the question.
> > We have added an ablation study to our revised paper, comparing DCD with the traditional contrastive decoding method. The results are shown below. This indicates that the answer confidence encapsulates valuable information about the relative informativeness of each modality for a given question. While confidence alone may not serve as a reliable indicator, the rich information it conveys can be leveraged to enhance overall performance.
> >
> > |                      | **ViQuAE** | **InfoSeek** |
> > |:--------------------:|------------|--------------|
> > | Contrastive Decoding | 70.10      | 49.05        |
> > | DCD                  | 76.49      | 54.90        |
> >
> > Regarding the **subtraction operation**, it plays a crucial role in reducing the influence of the conflicting modality. Moreover, the subtraction is scaled based on the confidence levels of each modality. This dynamic scaling allows the model to adaptively weigh the modalities according to their reliability at each inference step, resulting in more robust and contextually appropriate outputs.

---

> > > ### Author Response · Authors · 2024-11-21
> > > **Rebuttals (3/n)**
> > >
> > > 8. Q8. The explanation of why larger models benefit more from the prompting strategy is unconvincing. The assumption that incorrect predictions stem from cross-modality knowledge conflicts contrasts with the demonstration that such conflicts exist across various model scales. If these conflicts consistently occur across modalities, what enables larger models to implicitly discern which modality's knowledge is more reliable?
> > >
> > >
> > > Thank you for this insightful question. Please allow us to clarify the reasoning behind why larger models benefit more from the prompting strategy and the potential implicit rules of discerning correct modality knowledge.
> > >
> > > We introduce two prompting strategies: the first one only reminds the VLM of the existence of knowledge conflicts, while the second one gives the model both textual and visual answers and lets it choose from them.
> > > The intuition here is based on the observations of the textual and visual answers, which we include in the Appendix A.
> > > These cases indicate that the textual modality is better at memorizing date and location, while the visual components are better at memorizing the correlation between an entity and its names and the relationship among entities.
> > > Observations like these align with human’s cognition towards multimodal information that different modalities may be leveraged differently according to each of their advantages in presenting distinct information.
> > > Thus, the goal of our prompting strategies is to leverage the model's internal commonsense reasoning to evaluate and determine which modality's knowledge is more reliable.
> > >
> > >
> > > In the prompting strategy, the model has to determine which modality might contribute more relevant knowledge to the current questions.
> > > The differences lie in whether the knowledge from both modalities is implicit or explicit.
> > > The implicit knowledge means that the knowledge is not shown in inputs, while the explicit knowledge means that the knowledge is presented in the inputs.
> > > In the first setting, the knowledge is implicit.
> > > The process of recognizing conflicts and self-improve over the reasoned conflicts is all done by the model, which is obviously more difficult.
> > > Reflected on the results, the 7b model does not gain almost any improvement from it because of its relatively poor knowledge and reasoning ability.
> > > In the second setting, the knowledge is explicit. Provided with answers from both modalities, the model only needs to determine the more informative modality, which is an easier task compared to the first setting.
> > >
> > > Overall, larger models are inherently better at leveraging both prompting strategies due to their stronger reasoning and commonsense capabilities.
> > > The results reflect this distinction, and the differences in model performance under these strategies align with our observations and underlying assumptions.

---

> > > > ### Author Response · Authors · 2024-11-26
> > > >
> > > > Hi Reviewer TP54,
> > > >
> > > > Sincerely thanks again for your time and efforts in reviewing our submission.
> > > >
> > > > As the discussion period is ending soon, we wonder if there's any possibility that you can consider having a look at our reply. We would like to hear your feedback. If you have any other questions, please feel free to follow up anytime. We are committed to making any necessary revisions to further improve our work.

---

> > > > > ### Comment · Reviewer_TP54 · 2024-11-30
> > > > >
> > > > > Thanks to the author for the detailed rebuttal. While the response addressed several concerns, some critical issues remain unresolved.
> > > > >
> > > > > Q3: The author’s explanation regarding delta accuracy does not fully address my original question. Delta accuracy, defined as (RAcc_text - RAcc_vis), does not account for scenarios where both models produce incorrect predictions. For example, if the correct answer is A, but the LLM predicts B and the VLM predicts C, delta accuracy would be 0, while the failure rate (FR) would be 1. According to equation (5), the conflict rate (CR) would equal FR minus delta accuracy, which in this case would be 1. However, in this example, the incorrect predictions are more likely due to weak reasoning capability rather than knowledge conflict. This distinction requires further clarification.
> > > > >
> > > > > Q5: The author attributes knowledge conflicts (i.e., potentially inconsistent parametric knowledge) to the misalignment of dataset distributions caused by the separate training of the pre-trained LLM and visual encoder, as discussed in Section 3.1. This explanation suggests that such conflicts should affect all models equally, as the dataset misalignment is objective. The conclusion in Sec 4.3 (line 291,  regardless of the model’s scale and architecture, the likelihood of parametric knowledge conflicts remains relatively constant) also supports that. However, the results in Table 2 show that knowledge conflict levels vary with model scales. This inconsistency between the theoretical explanation and empirical results requires further elaboration.
> > > > >
> > > > > Q7: The ablation study provided by the author demonstrates the empirical advantage of DCD. However, the theoretical justification remains unconvincing. Section 5.1 demonstrates that confidence alone is not a reliable indicator of reasoning capability. Yet, in Section 6.1, confidence is used as a scaling factor for logits, influencing all logits used in the final reliability calculation. This approach appears counterintuitive, especially given that confidence has already been shown to be unreliable. A more robust theoretical explanation is needed to strengthen this argument.

---

### Author Response · Authors · 2024-11-21
**Summary of the Rebuttal**

Thanks to all the reviewers for their valuable suggestions and insightful questions!

Here we summarize the revisions in our paper according to the reviews and clarify the scope of our work.

## Revisions

1. We clarify which cases are taken into consideration in the Flip Rate(FR) in Section 3.2.2.
2. We explain the intuition and the calculation of the Conflict Rate(CR) in Section 4.2.
3. We report the $\Delta$Acc in Table 2 and clarify that we are reporting the lower bound of the CR in the table.
4. We add an ablation study on LLaVA-7b, comparing the proposed dynamic contrastive decoding (DCD) with the traditional contrastive decoding, in Appendix B.3.

## The scope of our work

Our work focuses on the cross-modality parametric knowledge conflicts in VLMs, more specifically, between the parametric knowledge of the language and the vision components in a single VLM. We aim to detect, interpret, and mitigate them at inference time. Knowledge conflicts and other VLM issues often entangle with each other during inference, making it challenging to isolate cases directly attributable to knowledge conflicts. To address this, we have done a lot of work to disentangle them by estimating the lower bound of the conflict rate and interpreting them with a contrastive metric. Consequently, our choice of evaluation datasets focuses on knowledge-intensive tasks, as they offer a cleaner and more controlled environment to focus solely on a model's knowledge capabilities. We deeply value the reviewers’ suggestion to explore more general datasets and have carefully considered them. However, these datasets typically require the model to exhibit a broader range of abilities, introducing additional complexities that could hinder the analysis of the cross-modality knowledge conflicts. To ensure the validity and clarity of our analysis, we believe it is essential to remain focused on knowledge-intensive datasets.

---

### Note · Authors · 2025-01-02

I have read and agree with the venue's withdrawal policy on behalf of myself and my co-authors.